# MONET: Multi-omic module discovery by omic selection

**Nimrod Rappoport**, **Roy Safra**, **Ron Shamir** *

The Blavatnik School of Computer Science, Tel Aviv University, Tel Aviv, Israel

* rshamir@tau.ac.il

**Data Availability Statement:** Digit dataset is available at: https://archive.ics.uci.edu/ml/machine-learning-databases/mfeat/. scNMT data are available at: https://github.com/BIRSBiointegration/Hackathon/tree/master/scNMT-seq. TCGA Breast

## Abstract

Recent advances in experimental biology allow creation of datasets where several genome-wide data types (called omics) are measured per sample. Integrative analysis of multi-omic datasets in general, and clustering of samples in such datasets specifically, can improve our understanding of biological processes and discover different disease subtypes. In this work we present MONET (Multi Omic clustering by Non-Exhaustive Types), which presents a unique approach to multi-omic clustering. MONET discovers modules of similar samples, such that each module is allowed to have a clustering structure for only a subset of the omics. This approach differs from most existent multi-omic clustering algorithms, which assume a common structure across all omics, and from several recent algorithms that model distinct cluster structures. We tested MONET extensively on simulated data, on an image dataset, and on ten multi-omic cancer datasets from TCGA. Our analysis shows that MONET compares favorably with other multi-omic clustering methods. We demonstrate MONET's biological and clinical relevance by analyzing its results for Ovarian Serous Cystadenocarcinoma. We also show that MONET is robust to missing data, can cluster genes in multi-omic dataset, and reveal modules of cell types in single-cell multi-omic data. Our work shows that MONET is a valuable tool that can provide complementary results to those provided by existent algorithms for multi-omic analysis.

This is a *PLOS Computational Biology* Methods paper.

## Introduction

Modern experimental methods can measure a myriad of genome-wide molecular parameters for a biological sample. Each type of such parameters is called "omic" and is measured by a different method. Analysis of omic data improved our understanding of biological processes and human disease, and is now used in therapeutic decisions [1]. While each experiment usually measures only one omic, several experiments can be performed on the same biological sample, resulting in multi-omic datasets. Large consortia such as TCGA and ICGC collected multi-omic data from tens of thousands of tumors [2,3]. Analysis of these data can further improve our understanding of cancer biology and suggest novel treatments.

cancer microarray data are available at: http://firebrowse.org/?cohort=BRCA&download_dialog=true. All other TCGA data are available at: http://acgt.cs.tau.ac.il/multi_omic_benchmark/download.html.

**Funding:** Study was supported in part by the Israel Science Foundation (grant 1339/18 and grant 3165/19 within the Israel Precision Medicine Partnership program), German-Israeli Project DFG RE 4193/1-1. NR was supported in part by a fellowship from the Edmond J. Safra Center for Bioinformatics, Tel Aviv University, and by the Planning and Budgeting Committee (PBC) fellowship for excellent PhD students in Data Sciences. The funders had no role in study design, data collection and analysis, decision to publish, or preparation of the manuscript.

**Competing interests:** The authors have declared that no competing interests exist.

Many algorithms have been developed in recent years to analyze multi-omic data, and most prominently, to detect subtypes of cancer, a task termed multi-omic clustering [4,5]. The vast majority of multi-omic clustering algorithms assume that a *common underlying structure* exists across all omics, and use all omic datasets to reveal this structure. Among the algorithms developed under this assumption are SNF and NEMO [6,7], as well as matrix factorization based methods such as MOFA+ [8], iClusterBayes [9] and MultiNMF [10]. However, this assumption does not always hold. For example, expression and mutation data do not seem to share the same structure. Even more closely related omics, such as expression and methylation, differ. This is demonstrated by the low agreement in clustering solutions that are produced based on different omics [11,12], and was also shown in a number of recent papers [13,14]. Moreover, in a recent benchmark we performed, we observed that solutions based on single omics can sometimes be more clinically relevant than solutions based on multiple omics [5]. Algorithms that can cluster patients while *accounting for the disagreement between omics* are therefore required.

Several recent methods addressed the distinct structure in different omics by using Bayesian statistics and modeling the different omics and their correlations. Savage et al. performed clustering on two omics, while allowing samples to be *fused* or *unfused* [15]. A fused sample belongs to a cluster spanning both omics, while unfused samples can belong to different clusters in the two omics. PSDF extended this framework to support feature selection [16].

MDI supports more than two omics [17]. Each omic has its own clustering, but clusters in different omics match each other. The probability that a sample will belong to matching clusters in two different omics has a prior that is higher the more these two omics are similar. In TWL [14], as in MDI, each omic also has its own clustering, and clusters in different omics match each other. A prior is placed such that samples are more likely to belong to the same cluster in different omics. BCC assumes a model with a global clustering and a clustering for each omic separately, and the global clustering serves as a Bayesian prior for each omic-specific clustering [18]. Clusternomics represents the global clustering as a Cartesian product of the omic-specific clusters, and can also map several such clusters into the same global cluster [13]. These methods have several limitations. MDI and TWL include only omic specific clusters, without providing a global clustering solution, and leave it to the user to choose between multiple clustering solutions. MDI, TWL and BCC further require that clusters in different omics match each other. Clusternomics' approach of representing global clusters as a Cartesian product of omic-specific clusters is less suited to find signals that are weak but consistent across many omics, and results in a high number of clusters. All methods except PSDF require a sample to belong to a coherent cluster in each of the omics, and PSDF is limited to only two omics. Furthermore, all available methods are based on Bayesian statistics, which requires explicit modeling of each omic, and is slow to optimize.

Here we present MONET (Multi Omic clustering by Non-Exhaustive Types), an algorithm for detection of patient modules for multi-omic cancer data. MONET uses ideas from MATISSE [19], an algorithm to detect gene modules, and generalizes its algorithmic approach to multi-omic data. In MONET's unique approach to multi-omic clustering, the goal is to form patient *modules*, such that each module can use only a *subset* of the omics. Thus, MONET can find patient modules with a common structure across some omics, and disregard other omics in that module, allowing different omic subsets for different modules. Note that this differs from ignoring an omic altogether, because an omic that is not used for one patient module can be used for other modules. MONET's solution allows outlier patients, who do not belong to any module.

We show that MONET finds biologically and clinically relevant patient modules in several datasets, giving results that compare favorably to those obtained from existent multi-omic

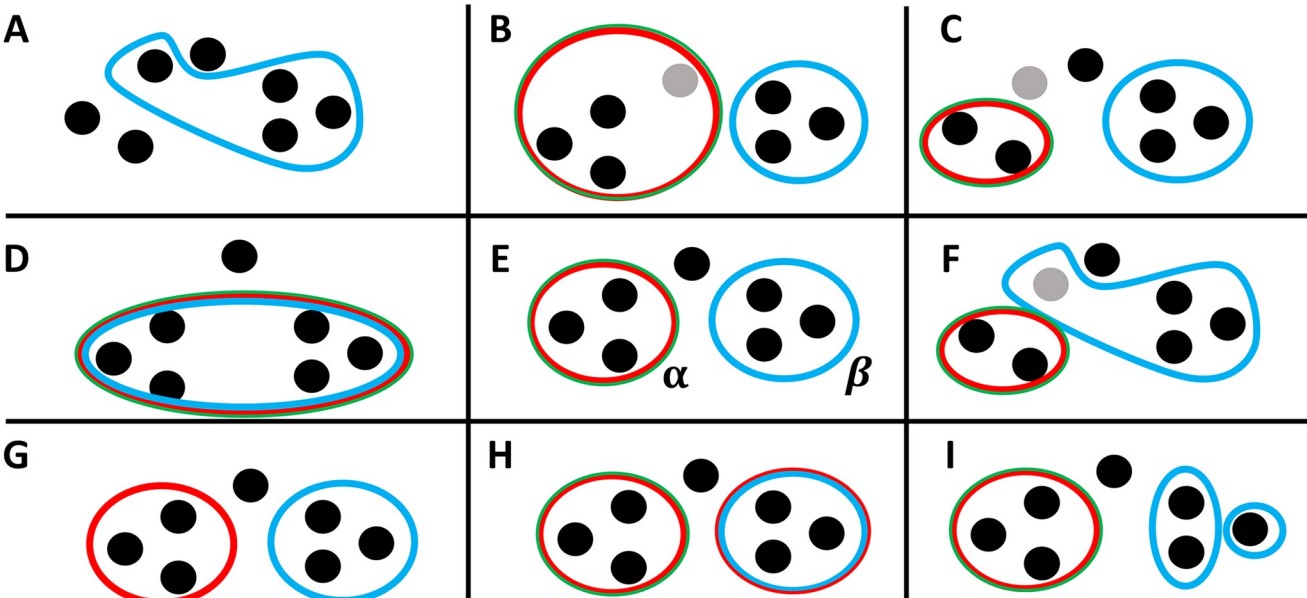

**Fig 1. Actions performed by MONET when detecting heavy modules. Dots represent samples, and enclosing circles represent modules.** The colors of the enclosing circle represent the omics covered by the module. Panel E shows the current state–two modules, where the left module ($\alpha$) is covered by two omics and the right module ($\beta$) by one. An additional sample is lonely, i.e., does not belong to any module. Each other panel shows one action. B: the grey sample is added to module $\alpha$. C: the grey sample is removed from module $\alpha$. F: the grey sample moves into module $\beta$. I: module $\beta$ is split. H: an omic is added to module $\beta$. G: an omic is removed from module $\alpha$. D: modules $\alpha$ and $\beta$ are merged. A: module $\alpha$ is discarded. In the shown case one of its samples is added to module $\beta$, and the other two become lonely. Actions for splitting module with omic or by adding omic are not shown.

clustering methods. Furthermore, we show that MONET is useful for other biomedical tasks, as it successfully finds modules of genes, and of cells in single-cell data.

## Methods

### Overview

The input to MONET is a set of $L$ omic matrices. Matrix $l$ has $n$ samples and $p_l$ features. The output is a set of modules, where each module is a subset of the samples. Modules are disjoint, and not all samples necessarily belong to a module. Samples not belonging to a module are called *lonely*. Each module $M$ is characterized by its samples, denoted *samples*($M$), and by a set of omics that it covers, denoted *omics*($M$). Intuitively, *samples*($M$) are similar to one another in *omics*($M$).

MONET works in two phases. It first constructs an edge-weighted graph per omic, such that nodes are samples and weights correspond to the similarity between samples in that omic. In the second phase, it detects modules by looking for heavy subgraphs common to multiple omic graphs.

### Omic graphs

MONET constructs a graph $G_l$ for each omic $l$ separately. $G_l$ is a full graph on $n$ nodes. Denote by $sim_l(u, v)$ some similarity measure between samples $u$ and $v$ in omic $l$. The weight assigned to edge $(u, v)$ in omic $l$, denoted by $w_l(u, v)$, is given by a function of the similarity between these two samples which we term "weighting scheme". This function is denoted $f$:

$$w_l(u, v) = f(sim_l(u, v))$$

The weight of a module is defined as:

$$weight(M) = \Sigma_{l \in omics(M)} \Sigma_{u,v \in samples(M)} w_l(u, v)$$

## The optimization problem

MONET's objective function is to find a disjoint set of modules $M_1, M_2 \ldots$ maximizing $\Sigma_i weight(M_i)$.

Importantly, we require that the weighting scheme returns values that are both positive and negative. High positive values indicate that the two samples are similar and should belong to the same module for omic $l$, while low negative values indicate the converse. A module with a positive weight therefore contains samples that are on average highly similar in the omics covered by the module. If all edge weights are positive, modules will always improve their scores by adding more samples and omics. Note that we present MONET here as a combinatorial optimization problem, but for some weighting schemes, the weight of each edge has a probabilistic interpretation. In such cases, the weight of a module is interpreted as the score for a log-likelihood ratio test for whether $samples(M)$ form a module on $omics(M)$, under the simplifying assumption that modules and sample pairs are independent. More details on this probabilistic formulation are in the appendix.

To construct the omics graphs, any weighting scheme can be used. The scheme we used here is as follows. We first apply NEMO [7], a multi-omic clustering algorithm we recently developed, to each omic separately $R$ times, each time on randomly selected 80% of the samples. We set $c_l^r(u, v)$ to 1 if samples $u$ and $v$ clustered together in the $r$'th run on omic $l$, and to 0 otherwise. Denote by $avg(c_l^r)$ the average value of the $c_l^r$ matrix, and by $R(u, v)$ the set of NEMO executions in which both $u$ and $v$ were sampled. We set $w_l(u, v) = mean_{r \in R(u,v)}(c_l^r(u, v) - avg(c_l^r)) - C$. The constant $C$ controls the balance between modules that cover one omic (higher $C$ value) and modules that cover multiple omics (lower $C$ value). Here we used $C = 0.2$ and $R = 100$. For the classification experiments we used a different weighting scheme, which is based on a Gaussian mixture model. Its full details are in the appendix.

## Heavy module detection

Given all the omic graphs, MONET now detects modules with high weight by maximizing the objective function $\Sigma_M weight(M)$. There is no constraint on the number of modules, or an upper bound on module sizes, so the weighting scheme must create both positive and negative edges, otherwise the trivial optimal solution is a single module containing all patients and covering all omics. The problem of detecting heavy subgraphs in this setting is NP-hard even for the case of a single graph [19]. We therefore developed an iterative greedy heuristic for detecting heavy modules. The algorithm is initialized with a set of modules termed seeds. After seed finding, at every iteration MONET considers several possible actions, described below, that can increase the objective function. It then performs an action that provides the greatest improvement.

■ Seed finding: Seeds are found iteratively. The first seed is determined by constructing a graph where edge weights are the sum of the edge weights in all individual omics, randomly selecting a first sample, and constructing a module containing all omics, which contains the first sample and its $k$ neighbors with highest positive edge weights. All samples that were assigned to a module are removed from the graph, and the next seed module is sought. The procedure ends once $S$ seeds were found. In this work we used $S = 15$ seeds for all datasets, and $k = floor\left(\frac{n}{15}\right)$.

■Optimization actions: Once a set of seeds is found, MONET improves the modules iteratively in a greedy manner. In each iteration, a module $M'$ is selected at random, and MONET calculates the gain in the objective function from a set of possible actions concerning the module. It then chooses the action with maximal gain. It stops when no action provides a gain in any module. The actions considered are (see **Fig 1**):

- Add a sample to $M'$. All lonely samples are considered. Since we observed that this action is commonly chosen in initial iterations when $S$ and $k$ are both small, we allowed up to 10 (or $\frac{n}{50}$ if $n>1000$) samples to be added in a single action, to reduce the number of iterations.

- Remove a sample from $M'$.

- Move sample from module $M'$ to another module, or move a sample from another module to $M'$. All possible samples and modules are considered. Similarly to adding samples, we allow up to 10 (or $\frac{n}{50}$ if $n>1000$) sample switches in a single action.

- Add an additional omic to a module. All omics are considered.

- Remove an omic from a module. All the covered omics of the module are considered.

- Merge modules $M'$ and $M''$. The set of samples for the new module is $samples(M')\cup samples(M'')$. The omics for the new module are one of the following: 1. $omics(M')\cup omics(M'')$ 2. $omics(M')\cap omics(M'')$ 3. $omics(M')$ 4. $omics(M'')$. All four options are considered.

- Split $M'$ into two modules. For this action, a graph is constructed with nodes $samples(M')$, and where the weight of the edge between $u$ and $v$ is $\Sigma_{l\in omics(M')}w_l(u, v)$. In this graph we find a heavy subgraph $M''$, and create two modules, $M''$ and $M\backslash M''$. The omics of both modules are $omics(M')$.

- Discard $M'$. Each sample $u$ in $M'$ is moved to the module $M''$ with the highest sum of weights from $u$ to $M''$ using $omics(M'')$. If all these sums are negative, $u$ is made lonely.

- Create a new module using all lonely samples. MONET finds a heavy subgraph in each omic separately, and a module is created from the heaviest subgraph found.

- Split $M'$ by adding an omic. For every omic $l\notin omics(M')$, MONET looks at the subgraph induced by $samples(M')$ on $G_l$, denoted $G_l[samples(M')]$, and detects in it a heavy subgraph. Denote the nodes of the heavy subgraph by $U$. We then split $M'$ into two modules. In one module the nodes are $U$, and the omics are $omics(M')\cup\{l\}$. In the second module the nodes are $samples(M')\backslash U$ and the omics are $omics(M')$.

- Split $M'$ with an omic. As in the previous action, a heavy subgraph with nodes $U$ is found in $G_l[samples(M')]$, but here for every $l\in omics(M')$. Two modules are constructed. In one the nodes are $samples(M')\backslash U$ and omics are $omics(M')$. In the other samples are $U$ and the only omic is $l$ that produced the heavy subgraph.

MONET uses a parameter $\eta$ for the minimum module size. Actions that reduce the number of samples below $\eta$ are not executed, and module splits are considered under this restriction. Here we used $\eta = \max\left(round\left(\frac{n}{30}\right), 10\right)$.

To find a heavy subgraph in a graph, we use a heuristic based on Charikar's 2-approximation to the problem of maximum density subgraph [20]. We iteratively find the node with lowest (weighted) degree and remove it from the graph, until no node is left. We then choose the heaviest of the sequence of subgraphs obtained during this process. The complexity of the heuristic on an $n$-node weighted full graph is $O(n^2)$.

The MONET algorithm is guaranteed to converge to a local maximum, because the sum of weights within all modules is increasing in each iteration. The algorithm stops when no action on any module improves the objective.

In each iteration, all actions that do not involve finding heavy subgraphs consider each edge in each of the omic graphs a constant number of times. The complexity of all these actions is therefore $O(\Sigma_l(n+|E_l|))$, where $E_l$ is the number of edges in $G_l$. The complexity of splitting a

module and of creating a new module involves finding a heavy subgraph and is thus $O(\Sigma_l(n+|E_l|)+n^2)$. For the last two actions, for the same reason, the same complexity is needed for each omic considered for the split, and the overall complexity is $O(L(\Sigma_l(n+|E_l|)+n^2))$, which is therefore the overall complexity of each iteration. For full graphs, this gives a worst case complexity of $O(L^2n^2)$. The space complexity is $O(Ln^2)$.

In a post-processing step we perform empirical significance testing to filter modules. Given a module, we sample 500 modules of the same size and omics, and only keep the module if its weight is in the highest 1%. In practice we only performed the testing for modules of minimal size ($\eta = 10$ here), as we never found larger non-significant modules. Samples that do not belong to any module after filtering are marked as lonely.

Since the algorithm for finding heavy modules is only guaranteed to converge to a local maximum, the algorithm is repeated multiple times, and the best solution is returned. Unless otherwise specified, we used 15 repeats for the analyses performed in this work.

## Additional MONET features

■Partial datasets: MONET can handle datasets where only a subset of the omics were measured for some samples. Such samples are added to all omic graphs, but in omics where these samples were not measured their nodes have no edges. This way, omics in which no data were measured for a sample do not affect the decision of assigning the sample to a module.

■Sample classification after clustering: Once modules were calculated from the data, MONET can naturally classify new samples into modules. For each module $M$, MONET calculates the gain in $weight(M)$ from adding the new sample $u$ to $M$: $\Sigma_{v\in samples(M)}w_l(u, v)$, and classifies the sample to the module with maximal gain. If the gain is always negative, the sample is not classified to any module. This computation takes $O(nL)$ given that the edge weights were already calculated.

## Testing methodology

We applied MONET and several other algorithms to simulated, image and cancer datasets that are described later. Here we outline the way we evaluated the results.

**Clustering assessment.**   To assess a clustering solution where the true clustering of the data is known, we used the Adjusted Rand Index (ARI) [21]. Note that the ARI can compare solutions with different number of clusters and different cluster sizes. On cancer datasets from TCGA we performed survival analysis to assess the distinction in survival between the different groups of samples, and tested enrichment of known clinical parameters. For the survival analysis we used a permutation-based approach to perform the log-rank test, since the widely used asymptotical version of this test tends to overstate significance, and specifically for TCGA data [22–24]. We also used permutation testing to assess the enrichment of clinical parameters [5]. The clinical parameters we considered were gender, age at diagnosis, pathological stage and pathologic M, N and T. In addition we considered known subtype definitions—PAM50 for breast cancer [25] and the French-American-British classification (FAB) for AML [26].

**Partial datasets experiments.**   For cancer datasets, we sampled 40% of the patients, partitioned them into three equal groups, and removed every group from one of the omics. For the image dataset we removed 20% of the samples in each omic independently. We then applied MONET to the data and calculated ARI with MONET's solution on all data. We repeated this experiment 10 times.

**Classification experiments.**   to perform experiments on a dataset we first applied MONET to it. Denote MONET's solution by $Sol_{all}$. We then partitioned the samples in the dataset into 10 equal folds. For every fold $i$, we applied MONET to all samples except those in

the fold, and denote the solution by $Sol_i$. We define the *stability* of the fold to be $ARI(Sol_{all}, Sol_i)$ where the ARI is computed using only samples that appear in both $Sol_{all}$ and $Sol_i$. We then classified the held out samples to the modules from $Sol_i$, and denote the solution after classification by $\hat{Sol_i}$. We define the *Rand Index following classification (RFC)* of the fold to be $ARI(Sol_{all}, \hat{Sol_i})$, where the ARI is now measured across all samples. For datasets where the ground truth is known we also measured $ARI(ground\_truth, Sol_i)$, and $ARI(ground\_truth, \hat{Sol_i})$, and term them the *pre-classification accuracy* (preCA) and *post-classification accuracy* (postCA) respectively.

**Simulations.**   The simulations are described in the appendix.

**Ovarian cancer analysis.**   To check the clustering solution for enrichment of clinical parameters we used chi-squared test for discrete features (e.g. tumor stage) and Kruskal-Wallis for numeric ones (e.g. age). We also used chi-square to test for enrichment of mutations and used Benjamini-Hochberg to correct for multiple hypotheses. To find genes and miRNAs that are highly expressed in a module, we performed a one sided t-test (with $\alpha = 0.05$) comparing the expression level in the module and the rest of the samples (after log normalization) and corrected for multiple hypotheses with Benjamini-Hochberg. Survival analysis was performed as described for the other TCGA datasets. To determine differential survival while controlling for the age and stage, we fitted a Cox multivariate proportional hazard model.

## Results

### Simulated datasets

We first performed two simulations to test MONET's approach to multi-omics clustering. In the first, we simulated 300 samples from five equal-size modules in two omics. Module 1 covers only the first omic, module 2 only the second omic, and modules 3–5 cover both omics (**Fig A in S1 Appendix**). We added five outlier samples that do not belong to any module. MONET correctly identified the modules (ARI = 0.92) and their corresponding omics (**Fig B in S1 Appendix**). In another experiment, we simulated 150 samples from five modules in three omics (**Fig C in S1 Appendix**). Module 1 covers all omics. Modules 2–5 cover all omics, are indistinguishable in omic 1, but belong to different clusters in omics 2 and 3. Only a small number of features separate the modules in omic 2, so the signal in omic 2 is weak. When presented with only omics 1 and 2, MONET identified module 1 but chose to treat modules 2–5 as one module that only covers the first omic (**Fig D in S1 Appendix**). When faced with omic 3 as well, the ARI equaled 1, and MONET identified these samples as coming from different modules that cover all omics (except for one module whose samples were very different in omic 2, which does not cover that omic) (**Fig E in S1 Appendix**). These simulations highlight MONET's approach to multi-omic integration, where sample modules can cover only a subset of the omics, based on the strength of the clustering structure in these omics. Full details on the simulations are in the appendix.

### Digits dataset

We next tested MONET in a dataset where the ground truth is known. The dataset [27] contains six types of features ("omics") of 2000 images of the handwritten digits 0–9. For most tests, we used 400 images. See additional details in the appendix.

We applied MONET and seven other methods to the data. We chose BCC, MDI, Clusternomics and TWL, which model disagreement between omics. We also chose SNF and NEMO to represent general multi-omic clustering methods. SNF is widely used, and we recently showed NEMO's high performance [7]. We also included MOFA+ [8], a widely used multi-

omic dimension reduction method. While MOFA was not developed specifically to cluster samples, its low dimensional representation can be used to cluster samples. Each method clustered the data into 10 groups. Note that MONET cannot get as input the number of modules, so we instead shifted the edge weights of the omic graphs to encourage about 10 modules (see details in the appendix). **Fig 2A** shows that MONET outperformed the other methods that model omic disagreement, and was comparable to SNF and NEMO. When ignoring lonely samples, MONET was slightly better than SNF and NEMO. Several modules found by MONET covered only a subset of the omics, suggesting a different structure in different omics (**Fig E in S1 Appendix**). Methods modeling omic disagreement were much slower than SNF, NEMO and MONET, which required a few seconds or minutes (**Fig 2B**).

In order to test MONET's scalability to thousands of samples, we also executed MONET on all 2000 images in the dataset. MONET took almost six hours to run, compared to less than ten minutes on 400 images. This was mainly due to increased runtime per iteration, but also because more iterations were required for convergence (**Fig G in S1 Appendix**). The performance was largely unchanged, with the ARI decreasing from 0.79 to 0.78.

## Cancer datasets

We next executed the same eight methods on real cancer datasets from TCGA, each containing three omics: mRNA expression, DNA methylation and miRNA expression. We used ten cancer types: Acute Myeloid Leukemia (AML), Breast Invasive Carcinoma (BIC), Colon Adenocarcinoma, Glioblastoma Multiforme (GBM), Kidney Renal Clear Cell Carcinoma (KRCCC), Liver Hepatocellular Carcinoma, Lung Squamous Cell Carcinoma (LUSC), Skin Cutaneous Melanoma, Ovarian serous cystadenocarcinoma and Sarcoma. Dataset sizes ranged from 170 to 621 patients. Full details on the datasets are available in our recent benchmark [5]. We used differential survival between clusters as an assessment criterion for the quality of a clustering solution (see Methods). MONET's modules for all cancer datasets are available in S1 Supporting Data.

As we can see in **Fig 2C**, MONET and NEMO had the highest number of cancer types with significantly different survival (at significance level 0.05), with 6 such types. MDI came next with 5, and the other methods had 3–4. Remarkably, in our recent benchmark, eight other multi-omic clustering methods, including the factorization-based methods iClusterBayes and MultiNMF, achieved significance for at most five cancer types. NEMO and MONET were also the best performers in terms of the number of subtypes with enriched clinical parameters (**Fig 2D**). The cancer types for which MONET and NEMO obtained a significant difference in survival were not identical. While both had different survival in AML, GBM, liver hepatocellular carcinoma and Sarcoma, NEMO found differential survival in BIC and melanoma, and MONET in KRCCC and ovarian cancer. Such a difference was also evident in the clinical parameters: NEMO found an enrichment in melanoma, while MONET found in LUSC. These results suggest that NEMO and MONET can be used complementarily. In terms of runtime, SNF and NEMO required seconds per dataset, MONET and MOFA+ a few minutes, and the remaining methods were an order of magnitude slower (**Fig 2E**).

The number of clusters chosen varied considerably among algorithms (**Fig H in S1 Appendix**). SNF had a mean of 2.8, TWL 3.4, NEMO, MONET and BCC 4–5, MOFA+ 5.7, MDI 8.9 and Clusternomics 26.5. The high numbers of MDI and Clusternomics are possibly due to attempting to model clustering in each individual omic. The log-rank p-value, number of enriched clinical labels, running time and number of clusters for each method and dataset are presented in **Tables A-D in S1 Appendix**.

MONET discovered modules that use different combinations of omics (**Fig 2F**). Most of the modules were based on only a single omic, and for several cancer types all modules covered

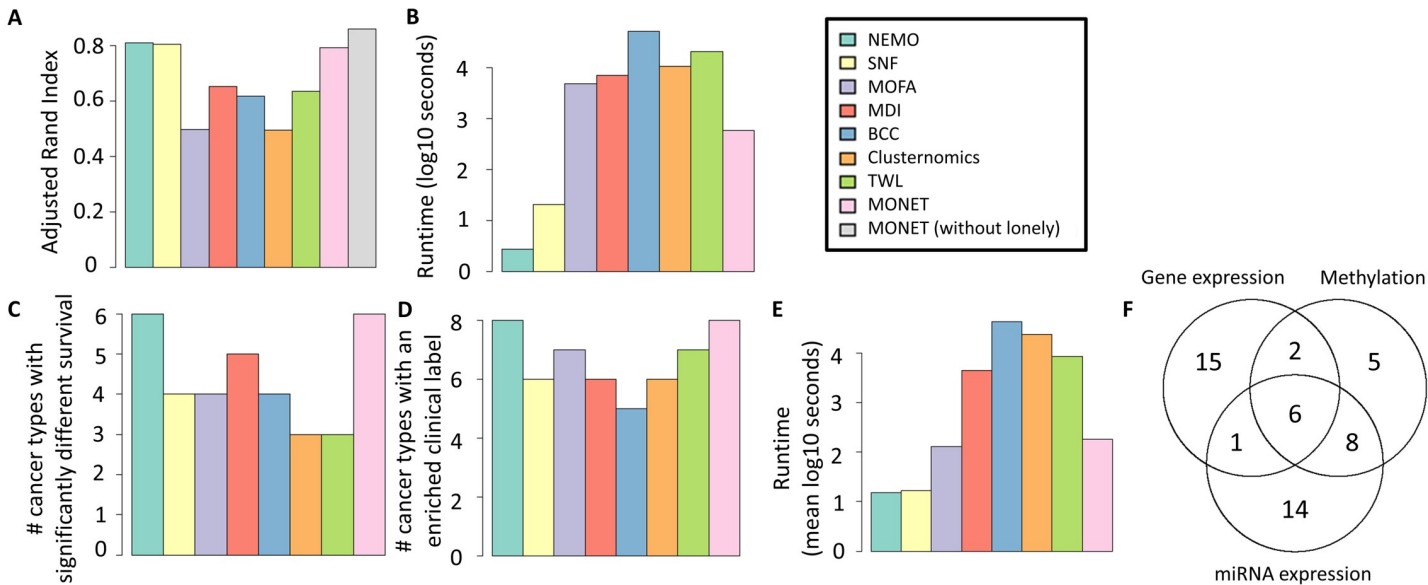

**Fig 2. Performance results.** A-B: Digits dataset. A: ARI of methods for multi-omic clustering. B: Run time. C-F: Results on ten TCGA cancer datasets. C: Number of cancer subtypes for which each method found a clustering with statistically different survival. D: Number of cancer subtypes for which each method found a clustering with an enrichment of a known clinical label. E: Run time. F: Number of MONET modules that cover each subset of omics.

only one omic. For some cancer types, this omic was the same for all modules, signifying a strong clustering structure in that omic. In none of the cancer types the solution contained only modules that covered all omics. These results suggest that different omics may have different structures, and that MONET reveals such differences. MONET also reported several (between 0 and 14) lonely samples per cancer (**Fig I in S1 Appendix**).

Since MONET is only guaranteed to converge to a local optimum, we experimented with using different numbers of restarts. In addition to the above results, which used 15 restarts, we also executed MONET with 1 and 50 restarts. For both 1 and 50 restarts, 6 cancer datasets were significantly associated with survival. The number of datasets with enriched clinical labels was 8 for one restart, and increased from 8 to 9 for 50 restarts, suggesting that MONET may benefit from more iterations. However, the clustering results of different restarts were generally similar to one another (**Fig J in S1 Appendix**).

## Additional analysis of the cancer results

We examined in more detail the clustering solution of MONET on the 287-patient ovarian cancer dataset. MONET found four modules in this dataset, with sizes 22, 63, 77 and 115, named M1-M4, and identified 10 samples as outliers (see **Fig K in S1 Appendix** for the feature heatmaps). While SNF and MDI seek to integrate structure across all omics (**Fig 3A**), MONET chooses the omics covered by each module. In its solution all modules cover the gene expression omic, and M1 also covers miRNA expression (**Fig 3B**). To assess the clinical relevance of MONET's modules, we examined the distribution of different clinical parameters across the modules. The modules showed significant differential survival (p = 0.036, **Fig 3C**), with M2 showing significantly better survival than the others (p = 4e-3). The modules showed differential survival even after correcting for age at diagnosis and clinical stage (p = 2e-4 using a Cox proportional hazards model). None of the other clustering algorithms found a solution with a significant difference in survival (**Fig 3D**). The modules were not significantly dependent of the clinical stage (0.056, chi-square test, 0.08 for Kruskal-Wallis), and they were enriched for

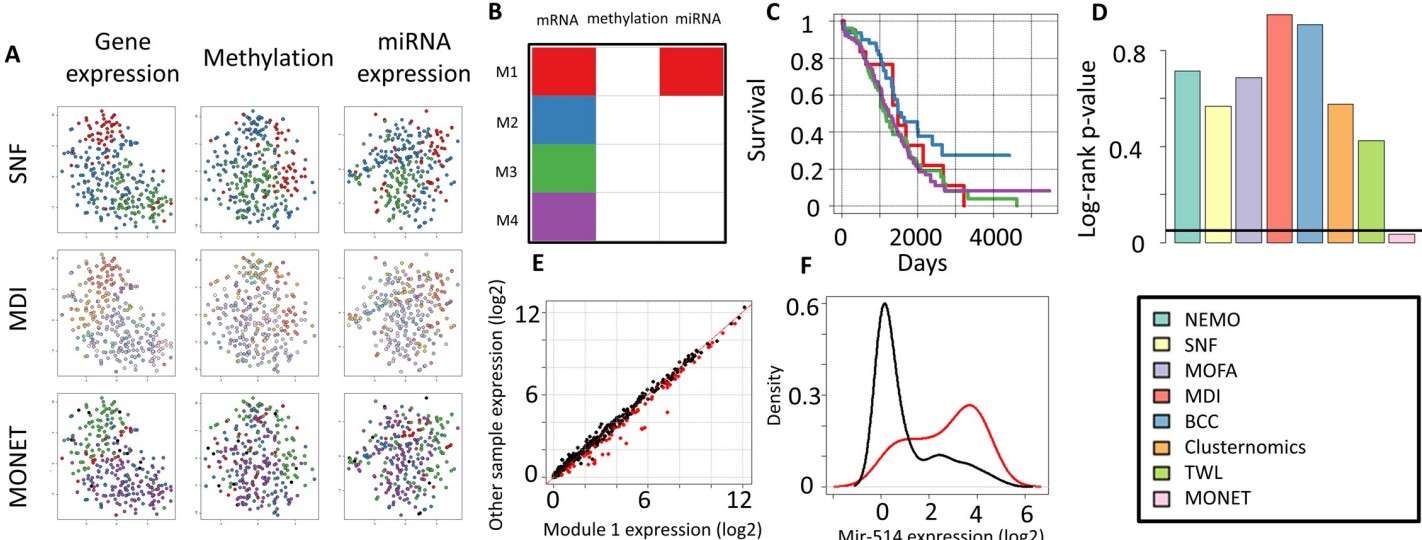

**Fig 3. Analysis of ovarian cancer.** A. t-sne [32] visualization of the solutions obtained by SNF, MDI and MONET on the data. Samples are colored by their assigned module. In MONET's panels, lonely samples are black. B. Omics covered by each MONET module. Columns are omics and rows are modules. C. Kaplan-Meier plot for the different MONET modules. D. p-value of the log-rank test for the clustering solutions of different methods. E. Comparison of miRNA expression for samples in MONET's Module 1 (x axis) and other samples (y axis). Genes that are significantly highly expressed in Module 1 are colored in red. F. Distribution of mir-514 expression in samples in Module 1 (red) and in other samples (black).

venous invasion status (8e-4, chi-square test, **Table E in S1 Appendix**) and for age at initial diagnosis (p = 7e-3 by Kruskal-Wallis, **Fig L in S1 Appendix**). No module was enriched for any mutation from a list of known driver mutations reported in TCGA's analysis of ovarian cancer [11] (see **Table F in S1 Appendix**).

We next characterized each module in more detail using clinical parameters and GO enrichment analysis (performed with Gorilla [28]) of differentially expressed genes with high module expression (see Methods). M1 had younger patients (p = 0.02, Wilcoxon test). It was the only module that included the miRNA omic. We found 21 miRNAs that were highly differentially expressed in M1's patients (**Fig 3E**, **Table G in S1 Appendix**), including mir-514, which was far higher on samples in M1 compared to all other samples (**Fig 3F**). It was recently reported to regulate proliferation and cisplatin chemoresistance in ovarian cancer [29]. M2 had significantly better survival, and its highly expressed genes were enriched for immune response. M3 was characterized by older samples (p = 4e-3, Wilcoxon test) without venous invasion (p = 2e-4, chi-square), and upregulation of genes involved in microtubule-based process (e.g. TUBB2B, TUBB4A). Finally, samples in M4 were enriched for venous invasion (p = 0.02, chi-square) and high expression of immune response and extracellular matrix organization related genes (e.g. MMP9 and multiple collagen subunits).

To understand the differences between M2 and M4, we found genes differentially expressed between them. M4 had higher expression of genes related to cell adhesion (e.g. collagen subunits), extracellular matrix (ECM) organization, and regulation of developmental process (e.g. WNT7A, WNT7B). Both the extracellular matrix and WNT signaling were previously reported to regulate ovarian cancer progression [30,31], and may explain the difference in venous invasion and survival between the modules. The high expression of ECM proteins may link M4 with the previously reported Mesenchymal subtype [11].

We also executed NEMO and MONET on each individual omic in the ovarian cancer data. MONET found a significant separation in survival for each omic individually (p-value 0.04–

0.05 in all omics), while NEMO did not find such separation for any. This shows MONET's effectiveness as a single-omic clustering approach (in this setting it is very similar to Matisse).

We observed that in several cases MONET used omics that were especially relevant for a specific dataset. For example, MONET's solution on GBM used only methylation in all modules. We executed spectral clustering and NEMO on each GBM omic separately and both algorithms found a solution with significant difference in survival only for the methylation dataset (p-value < 0.001 in both cases). Note however that MONET's solution often uses multiple omics (see **Fig 2F** for all cancer datasets and **Figs M-N in S1 Appendix** for the solutions on BIC and Sarcoma).

One of the main advantages of Bayesian methods is that they associate a posterior probability for each sample to belong to each cluster. MONET also provides a quantitative measure for the association between a sample and a module: the sum of weights between the sample and all the module's samples across all omics covered by the module. A similar association score can also be calculated for each omic separately (see **Fig O in S1 Appendix** for the scores for the ovarian cancer dataset). These scores can assist in better understanding of the data, on top of the binary module memberships. For example, **Fig O in S1 Appendix** shows that M1 has a weak structure in both the omics it covers, while the three other modules differ greatly in gene expression. The score also suggests that M3 samples have some similarity in methylation, as is also suggested by **Fig K in S1 Appendix**, though this level of similarity is not sufficient for M3 to cover the methylation omic. These observations appear consistent with the t-SNE plot for the data (**Fig 3A**).

## Partial datasets

Often in multi-omic datasets, some samples have measurements for only a subset of the omics. Such datasets are called *partial*. MONET can address such datasets by assigning edge weight 0 to samples in the omics that were not measured. We tested this ability using the Sarcoma dataset, which had modules covering all omics, and using the digits dataset. In each dataset we randomly removed samples from some omics (see Methods), applied MONET, and compared its solution to the solution using all samples, and to the ground truth in case of the digits dataset. The results are presented in **Fig 4A** and **Fig 4B**.

MONET's output on the digits dataset was quite robust, with only a slight deterioration in performance. The Sarcoma results were stable as well, but the stability highly varied between the omics from which samples were removed. Samples removed from the gene expression omic had lower ARI compared to samples removed from other omics, possibly indicating that MONET's solution is highly affected by that omic for that dataset. The ARI slightly differed for samples in the digits dataset as well depending on the omic from which they were removed (**Fig P in S1 Appendix**). These results suggest that MONET can be robustly applied to partial datasets.

## Classification

Given a clustering solution, MONET supports classification of new samples into modules (see Methods). We tested MONET's robustness and classification on the Sarcoma and digits datasets. For each dataset we performed an unsupervised version of 10-fold cross validation. We define the *stability* of a fold as the ARI between MONET's solution on all samples and MONET's solution for the current fold (which excludes 10% of the samples). We define the *Rand Index following classification* (RFC) of a fold as the ARI between MONET's solution on all samples and its solution on the fold following the classification of the 10% held out samples (see Methods). For the digits dataset, we also compared the result of every fold to the ground truth,

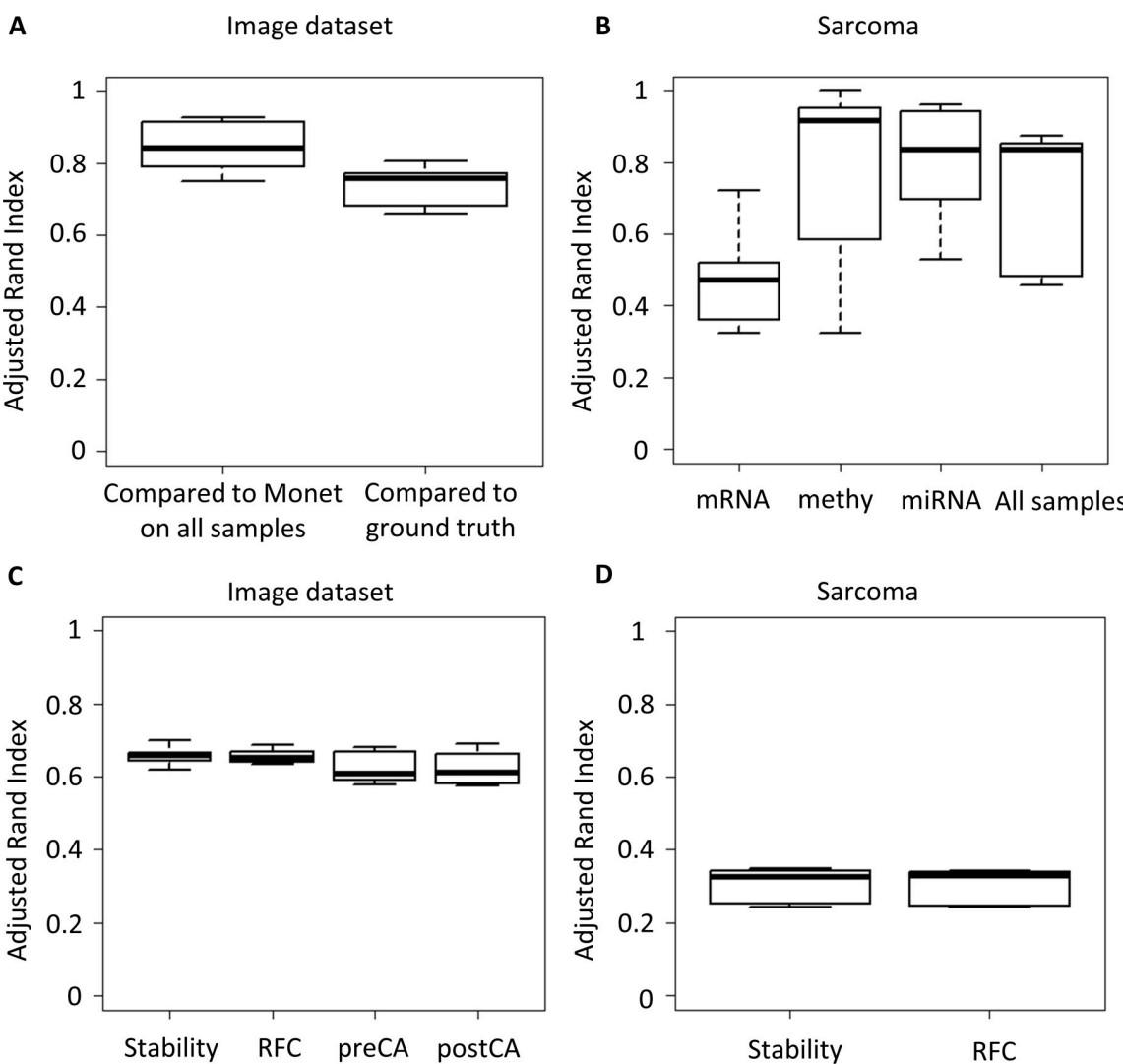

**Fig 4. Performance of MONET on partial datasets and in classification.** A. ARI on a partial version of the digits dataset compared to its solution on the full dataset and to the ground truth. B. ARI on a partial version of the Sarcoma dataset compared to its solution on all samples. Shown is the ARI for samples that were dropped from each one of the omics (three left boxplots), and for all the samples in the dataset (rightmost boxplot). C. Performance in classification experiments on the digits dataset. See Methods for the assessment criteria. D. Performance in classification experiments on the Sarcoma dataset. All boxplots are distributions over 10 random runs.

with and without the 10% of held out samples, and term them the *pre-classification accuracy* (preCA) and *post-classification accuracy* (postCA). Note that we used here the Gaussian mixture weighting scheme (which is described in the appendix), as in order to perform classification MONET calculates the edge weights for the new samples.

The results are presented in **Fig 4C** and **Fig 4D**. In the runs on the digits dataset, both the stability and RFC are high. 45 (11%) of the images were not classified to a module, as no module with positive classification score was found for them. In the runs on the Sarcoma dataset the results are only moderately stable, but the RFC is as high as the stability. This suggests that the classification is accurate, and that decrease in performance stems largely from the different clustering structure that is obtained from sampling the datasets. All samples were classified in this dataset. Overall, these results show that MONET's framework can be used to perform classification given new samples.

### Other biological tasks: Gene and single cell clustering

We next tested MONET on additional biological tasks. We used MONET to cluster 1532 genes measured by both RNA-seq and microarrays of the BIC TCGA dataset that exhibited high variance in both these omics. We used BIC because of its large sample size, and to demonstrate MONET's utility for in-depth analysis on an additional cancer type. MONET reported five main gene modules (**Fig 5A**, **Fig Q in S1 Appendix**). We used Gorilla [28] to perform enrichment analysis for these gene modules. Reassuringly, we found enrichment of biological processes that vary across breast cancer patients in several modules, including "mitotic cell cycle process", "immune system process", and "extracellular matrix organization". As expected, all gene modules covered both omics.

Finally, we applied MONET to single-cell data. Argelaguet et al. recently developed scNMT, a method that measured gene expression, DNA methylation and DNA accessibility at single cell resolution, and applied it to mouse embryos at embryonic days 4.5–7.5 [33]. We applied MONET to the gene expression and promoter methylation data of 619 single cells (**Fig 5B and 5C**). The modules obtained were highly enriched for specific cell types and embryonic days of development (**Tables H-J in S1 Appendix**). Several modules, across different cell types and stages of development, covered both omics, reflecting the widespread changes in expression and methylation during the onset of gastrulation [34,35]. Other modules used only gene expression, suggesting an overall stronger distinction between cell types at the expression level. One module covered only DNA methylation. This module comprised cells from different cell

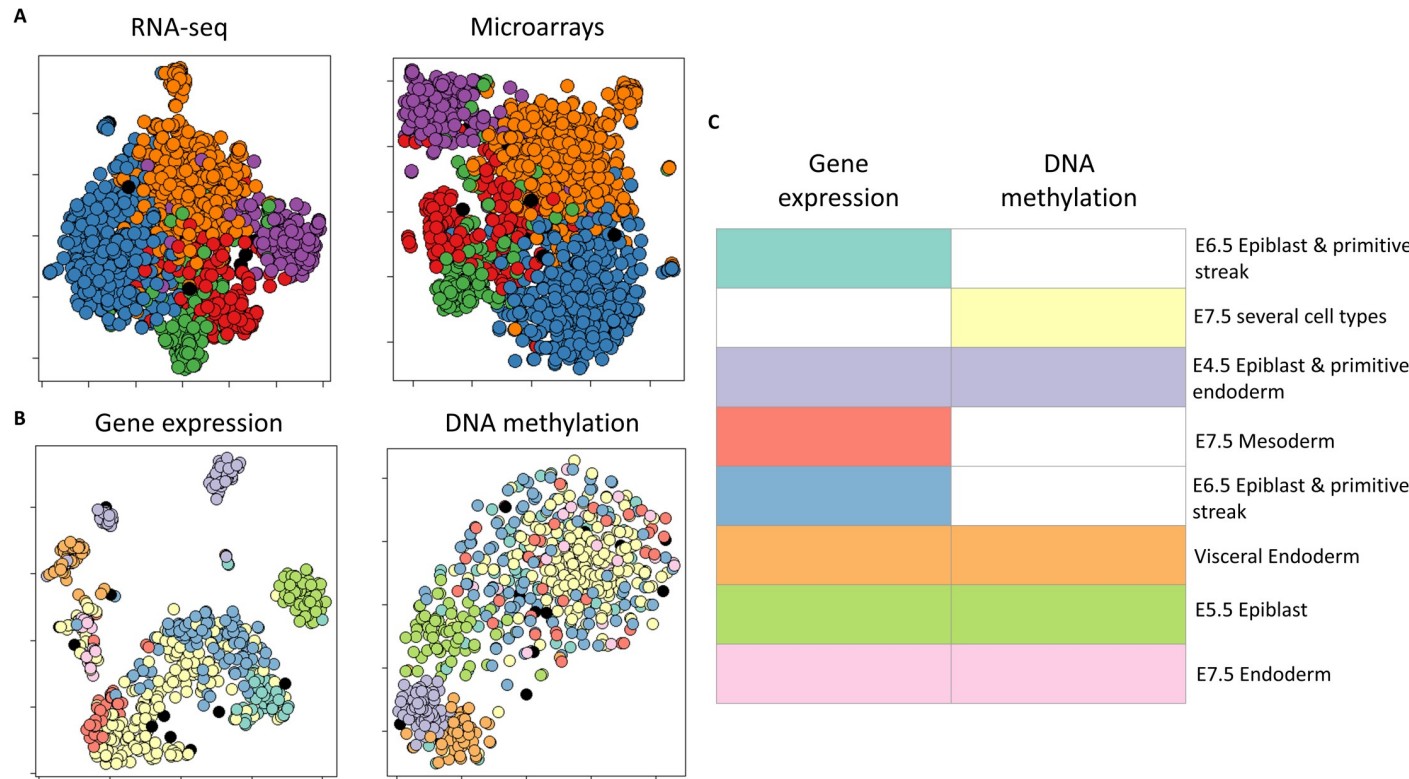

**Fig 5. Using MONET to cluster genes and single cells.** A. Gene clustering. t-sne visualization of MONET's gene modules on the BIC dataset. Genes are colored by MONET's output. Lonely samples are colored in black. B-C. Single cell clustering based on gene expression and DNA methylation of promoters, using the scNMT mouse embryonic development dataset. B. Like A, for MONET's solution on the dataset. C. Module omics identified by MONET. Rows represent modules and columns correspond to omics. Colored panels indicate that the module covers the omic.

types at E7.5, including cells from all germ layers, again highlighting that while the transcriptional signatures of different cell types differ at that stage, the promoter methylation profile of the different germ layers is still quite similar [33]. Overall, these results demonstrate that MONET can be applied and lead to insights in diverse biological scenarios.

## Discussion

We presented MONET, a novel multi-omic clustering algorithm. MONET can identify modules with structures present in some of the omics, without imposing these structures on other omics. MONET can also identify samples that do not fit any detected module. State-of-the-art methods that seek clusters across all omics often perform quite well, and structures that span all omics have been observed in many studies. We view these approaches as complementary to MONET, and suggest using both for multi-omic analysis. That is, data analysis can benefit from using both MONET as well as other algorithms that seek a common structure, and each of these approaches will reveal different aspects of the data.

It is challenging to interpret omics data and its clustering in the face of disagreement between omics. From a data analysis point of view, as we noted before, one can use different tools for the analysis. Methods that assume agreement between omics can be used, together with different formulations for omic disagreement: omic-specific clusters, omic-specific deviations from a global clustering solution, or clusters that apply in only a subset of the omics. From a biological point of view, a different structure between omics can reveal insight on biological regulation and disease. For example, for biological regulation, it is interesting to discover gene modules that are co-expressed but are not highly correlated on the protein level. As another example, in disease, the GBM G-CIMP subtype is associated with IDH mutations and a characteristic methylation phenotype, while its expression profile does not define the subtype as distinctly [36].

The edge weighting in MONET's omic graphs can be done by schemes tailored to the omic and data, allowing flexibility in the analysis. The weighting schemes used here to cluster patients, genes, and single-cells show MONET's ability in different biomedical domains. The weighting scheme can also shift the balance between modules with single or multiple omics, or place more emphasis on one particular omic.

Most multi-omic analysis methods assume that samples are present in all omics. This is rarely the case in datasets available today, such as TCGA. It is also likely that partial datasets will be prevalent in single-cell analysis, where measuring multiple omics from a cell is just beginning and is experimentally challenging. MONET's ability to analyze partial datasets will make it valuable in this setting.

MONET has several limitations. Using different weighting schemes allows flexibility, but it can be challenging to choose one that balances finding omic-specific signals and signals reinforced by different omics. The optimization problem MONET solves is NP-hard, so the algorithm is heuristic. Adding new actions to MONET's heavy subgraph algorithm can improve its output. While MONET is faster than methods modeling disagreement between omics and can easily be run on today's datasets, which contain hundreds of samples, it is currently not scalable to more than a few thousand samples. Future work can improve MONET's runtime, for example by removing edges in the omic graphs, or by discretizing the edge weights, which allows a more efficient implementation of Charikar's algorithm. The potential of MONET for classification warrants further validation in the cancer context. Finally, as MONET does not model the features in the dataset, understanding the molecular differences between modules requires additional analysis.

## Code availability

Code for MONET and for reproducing all results in this paper is in Github: https://github.com/Shamir-Lab/MONET.

## Supporting information

**S1 Appendix. Additional implementation details, and supporting figures and tables.** (DOCX)

**S1 Supporting data MONET's clustering results on the TCGA and scNMT datasets.** (ZIP)

## Acknowledgments

The results published here are based upon data generated by The Cancer Genome Atlas managed by the NCI and NHGRI. Information about TCGA can be found at http://cancergenome.nih.gov. The contribution of N.R. is part of Ph.D. thesis research conducted at Tel Aviv University.

## Author Contributions

**Conceptualization:** Nimrod Rappoport, Ron Shamir.

**Formal analysis:** Nimrod Rappoport, Roy Safra.

**Funding acquisition:** Ron Shamir.

**Methodology:** Nimrod Rappoport, Roy Safra.

**Project administration:** Ron Shamir.

**Software:** Roy Safra.

**Supervision:** Ron Shamir.

**Visualization:** Nimrod Rappoport, Roy Safra.

**Writing – original draft:** Nimrod Rappoport, Ron Shamir.

**Writing – review & editing:** Nimrod Rappoport, Ron Shamir.

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
