## [Decision Letter · Decision Letter 0]

1 Jul 2020

Dear Prof. Shamir,

Thank you very much for submitting your manuscript "MONET: Multi-omic module discovery by omic selection" for consideration at PLOS Computational Biology. As with all papers reviewed by the journal, your manuscript was reviewed by members of the editorial board and by several independent reviewers. The reviewers appreciated the attention to an important topic. Based on the reviews, we are likely to accept this manuscript for publication, providing that you modify the manuscript according to the review recommendations.

Sincerely,

Teresa M. Przytycka

Associate Editor

PLOS Computational Biology

Jason Papin

Editor-in-Chief

PLOS Computational Biology

[LINK]

Reviewer's Responses to Questions

**Comments to the Authors:**

Reviewer #1: The clarity of the exposition has been substantially improved, and the authors have responded to all of my points adequately.

There are only a couple of outstanding issues:

- Unlike other methods, MOFA+ is not mentioned in the Introduction.

- I still believe there is an overstatement of the meaning of the results (pg. 8): "different omics do have different structures". This refers to MONET's solutions on the cancer data, but MONET is designed to capture such signal. In the absence of a ground truth, I do not believe that such a strong statement can be made: it is possible that MONET's increased clustering flexibility is leading it to overfit and identify spurious correlations resulting from the choice of similarity function and optimisation objective.

These results suggest that different omics may have different structures.

- Regarding the significance of the survival curves of Figure 3C, it seems that M2 is the sole driver of that statistical significance. Without it, the MONET's modules would not show differential survival. Again, it seems that the text is overly optimistic.

Reviewer #2: I appreciate the authors' effort to revise and improve this manuscript. The presentation has been strengthened. The authors also added more analysis. Overall, this is a good method contribution with meaningful applications to cancer genomic data and potentially other multi-omics datasets.

I only have one remaining question that I hope the authors can clarify with either more comparison or at least more method comparison discussions.

It appears that the problem can also be approached by using non-negative tensor factorization. As a special case, this will be non-negative matrix factorization where there are quite a few earlier work in trying to identify the latent variables and also the hidden structures for the clusters, e.g., Zhang et al. (PMID: 22879375) Liu et al. (PMID: 24491042) and Shen et al. (PMID: 19759197). I suggest that the authors put MONET in the context of these related prior work and clarify the advantage.

**Have all data underlying the figures and results presented in the manuscript been provided?**

Reviewer #1: Yes

Reviewer #2: Yes

PLOS authors have the option to publish the peer review history of their article (what does this mean?). If published, this will include your full peer review and any attached files.

Reviewer #1: No

Reviewer #2: No
---

## [Editor Report · Decision Letter 1]

22 Jul 2020

Dear Prof. Shamir,

We are pleased to inform you that your manuscript 'MONET: Multi-omic module discovery by omic selection' has been provisionally accepted for publication in PLOS Computational Biology.

Best regards,

Teresa M. Przytycka

Associate Editor

PLOS Computational Biology

Jason Papin

Editor-in-Chief

PLOS Computational Biology

---

## [Editor Report · Acceptance letter]

9 Sep 2020

PCOMPBIOL-D-20-00697R1 

MONET: Multi-omic module discovery by omic selection

Dear Dr Shamir,

I am pleased to inform you that your manuscript has been formally accepted for publication in PLOS Computational Biology. Your manuscript is now with our production department and you will be notified of the publication date in due course.

With kind regards,

Sarah Hammond
